# Heterologous Overexpression of *ZmHDZIV13* Enhanced Drought and Salt Tolerance in *Arabidopsis* and Tobacco

Fang Wang [1,2,†] , Huiping Yan [1,†], Peng Fang [1,†], Xiangzhuo Ji [1] and Yunling Peng [1,2,*]

1    College of Agronomy, Gansu Agricultural University, Lanzhou 730070, China
2    State Key Laboratory of Aridland Crop Science, Gansu Agricultural University, Lanzhou 730070, China
*    Correspondence: pengyunlingpyl@163.com
†    These authors contributed equally to this work.

**Abstract:** The homeodomain leucine zipper (HD-Zip) IV transcription factor is indispensable in the response of plants to abiotic stress. Systematic studies have been carried out in *Arabidopsis*, rice and other species from which a series of stress resistance-related genes have been isolated. However, the function of the HD-Zip IV protein in maize is not clear. In this study, we cloned the HD-Zip IV gene *ZmHDZIV13* and identified its function in the stress response. Our phylogenetic analysis showed that *ZmHDZIV13* and *AtHDG11* had high homology and might have similar functions. The heterologous overexpression of *ZmHDZIV13* in *Arabidopsis* resulted in sensitivity to abscisic acid (ABA), salt tolerance during germination and drought tolerance in seedlings. Under drought stress, the transgenic *Arabidopsis* showed stronger drought resistance than the wild-type (control). The malondialdehyde content of *ZmHDZIV13* transgenic plants was lower than that of the control, and the relative water content and proline content were significantly higher than those of the control. After the drought was relieved, the expression levels of stress-related genes were up-regulated in transgenic *Arabidopsis*. These results show that *ZmHDZIV13*, as a stress-responsive transcription factor, plays a role in the positive regulation of abiotic stress tolerance and can regulate an ABA-dependent signaling pathway to regulate drought response in plants.

**Keywords:** homeodomain leucine zipper; *ZmHDZIV13*; maize; drought and salt tolerance; ABA

## 1. Introduction

Plant growth and development is a complex and diverse biological process. This development process is often affected by various biotic and abiotic stress factors, such as pathogens, low temperature, drought and salinity. In response to these stresses, plants induce various regulatory mechanisms occurring at cellular, molecular, physiological and biochemical levels [1]. The earliest abiotic stress signals that plants perceive and transmit are through molecular signaling pathways affecting the expression levels of a series of associated genes [2]. Among them, transcription factors play an important regulatory role in plant stress response. A transcription factor, also known as a transacting factor, is a protein molecule with a special structure that regulates gene expression. Some transcription factors are related to plant resistance. When plants are subjected to abiotic stress, a series of signal transmissions stimulate the expression of transcription factors. The stimulated transcription factors combine with corresponding cis-acting elements to inhibit or enhance gene expression and its downstream effects. In recent years, hundreds of transcription factors related to plant stress resistance (e.g., drought, high salt, low temperature), growth and development have been discovered one after another [3].

The homeodomain leucine zipper (HD-Zip) protein is an important transcription factor unique to plants. It has been reported in *Arabidopsis* [4], sunflower [5], rice [6], tomato [7], maize [8], alfalfa [9], wheat [10], barley [11], potato [12], cucumber [13], *Populus simonii* × *P. nigra* [14] and mulberry [15]. The HD-Zip protein includes two important functional

domains, the HD domain for DNA binding and a closely connected ZIP domain that is related to protein dimerization. There are 48 HD-ZIP family proteins in the *Arabidopsis* genome, and they are divided into four subfamilies (HD-Zip I-IV) according to DNA binding specificity, physical and chemical properties and other functional domains [16]. Related studies have shown that HD-Zip IV participates in root development and epithelial cell differentiation during plant growth and development while regulating the accumulation of anthocyanins and the formation of trichomes [17]. In addition, the HD-Zip IV family has a START domain, which plays an important regulatory role in response to drought and other abiotic stresses [18]. At present, multiple HD-Zip IV genes have been cloned from *Arabidopsis thaliana* [19], *Oryza sativa* [20], *Zea mays* [21,22], *Triticum aestivum* [23], *Nicotiana tabacum* [24] and *Cucumis sativus* [25]. For example, the *AtHDG11* (*HOMEODOMAIN GLABROUS11/ENHANCED DROUGHT TOLERANCE1*) gene from *A. thaliana* is an important member of the HD-START (HD-Zip IV) transcription factor family. The expression of this gene in plants can effectively promote root elongation and stomatal closure, thereby significantly increasing plant resistance to drought [26]. Cao et al. [27] transferred *At HDG11* into tall fescue (*Festuca elata*) and the overexpression of the gene enhanced drought and salt tolerance in transgenic plants. Wei et al. [28] constructed and transformed an *Arabidopsis At HDG11* overexpression vector into ryegrass (*Lolium perenne*) and found that compared with wild-type (WT) plants, the transgenic plants had lower levels of malondialdehyde (MDA) under drought stress and maintained high levels of superoxide dismutase (SOD) and catalase (CAT) activity. Yu et al. [29] also found that the overexpression of *At HDG11* in cotton (*Gossypium* spp.) and *Populus tomentosa* produced strong drought and salt tolerance, and the transgenic plants had more developed root systems. Li et al. [30] found that the overexpression of *AtHDG11* enhanced drought resistance in wheat (*Triticum aestivum* L.), and the transgenic plants had low stomatal density, low water-loss rate, high proline content and greater accumulation and activity of catalase and superoxide dismutase compared with WT plants. Zhu et al. [31] found that the overexpression of *AtEDT1/HDG11* in Chinese kale (*Brassica oleracea* var. *alboglabra*) enhances drought and osmotic stress tolerance, and the content of proline and the activity of active oxygen-scavenging enzymes in transgenic Chinese kale leaves were significantly higher than in wild Chinese kale. Banavath et al. [32] found that overexpression of *At HDG11* improved drought resistance and salt tolerance in transgenic peanut plants (*Arachis hypogaea* L.) by increasing free proline content, water-use efficiency (WUE), chlorophyll content and photosynthetic rate and decreasing stomatal density. Guo et al. [33] showed that *AtEDT1/HDG11* enhanced drought resistance by regulating stomatal density and WUE of *A. thaliana*. Fang et al. [34] isolated *ZmHDZIV14* from the HD-ZIP IV transcription factor family of maize and observed high homology between *ZmHDZIV14* and *At HDG11*. They found that heterologous overexpression of *ZmHDZIV14* in *A. thaliana* led to increased sensitivity to ABA, salt and mannitol tolerance during seed germination.

Initially, we conducted a BLAST search for homologous protein sequences in a corn database (Maize GDB) and found that *AtHDG11* (GenBank No. NM_105996) in *Arabidopsis* and *ZmHDZIV13* (GenBank No. BK008038) in maize are in the same clade with a degree of similarity of 53.51%, indicating high homology. We speculated that *ZmHDZIV13* has a similar function to the *AtHDG11* gene. At present, no research has been reported on the *ZmHDZIV13* gene in maize, and thus, its biological function remains unclear. In this study, we cloned the maize HD-Zip IV gene *ZmHDZIV13* (BK008038) and the function was identified. We found that stress induced by drought, salt (NaCl), ABA and mannitol treatments increased the expression of *ZmHDZIV13*. Overexpression of *ZmHDZIV13* caused ABA hypersensitivity and osmotic stress during the germination period of *Arabidopsis*. Moreover, compared to that of the WT, overexpression increased tolerance to drought stress. These results show that *ZmHDZIV13*, as a stress-responsive transcription factor, plays a role in the positive regulation of abiotic stress tolerance and is of great significance for improving maize stress resistance and enriching resources based on maize resistance genes.

## 2. Materials and Methods

### 2.1. Plant Materials

The experiment was carried out in the Gansu Key Laboratory of Arid Land Crop Science, Gansu Agricultural University. WT *Arabidopsis* seeds were obtained from this laboratory, tobacco (*Nicotiana tabacum*, T12) seeds were provided by the Gansu Academy of Agricultural Sciences and maize inbred lines (*Z. mays*, Zheng 58) were provided by Pingliang Breeding Station in Gansu Province.

### 2.2. Bacterial Strain, Plasmid and Transformation

Total RNA was extracted from the immature tassels of the maize inbred line Zheng 58, reverse-transcribed into cDNA as a template and PCR-amplified to obtain the full-length cDNA fragment (2372 bp) of *ZmHDZIV13*. The fragment was recovered and ligated into the pUCm-T-Easy vector. The complete sequence obtained by sequencing verification is stored in GenBank under the accession number BK008038.

The primers were designed to amplify the target gene fragment, *ZmHDZIV13*. We introduced an *Xba* I restriction site at the 5′ end of the corresponding target gene fragment and a *Sma* I restriction site at the 3′ end to connect with the pUCm-T cloning vector. Then, the target gene fragment was obtained by double digestion of *Xba* I and *Sma* I and connected with a 9580 bp fragment of pCAMBIA3300-Ubi-PRO II MCS-*bar* to obtain the pCAMBIA3300-35S-*ZmHDZIV13-bar* plant expression vector. The constructed plant expression vector pCAMBIA3300-35S-*ZmHDZIV13-bar* was transformed into *Agrobacterium tumefaciens* LBA4404, a single colony of bacteria was selected and the *Agrobacterium* plasmids were extracted for PCR identification. The plant expression vector was identified by double enzyme digestion of *Sma* I and *Xba* I. Finally, *A. tumefaciens* LBA4404 containing this vector was screened for and introduced into wild *A. thaliana* and wild tobacco by the flower-soaking method [35]. Homozygous transgenic lines from the T3 generation were used for further assays.

### 2.3. Measurement of Germination Rate and Root Traits and Evaluation of Drought Resistance

Thirty *Arabidopsis* T3 seeds were selected from each WT and transgenic lines, surface-disinfected and treated with different concentrations of ABA (0, 0.2, 0.4, 0.6 and 0.8 μM), salt (0, 50, 100, 150 and 200 mM) and mannitol (0, 50, 100, 150 and 200 mM) in their MS growth medium. The seeds were cultivated at 4 °C for three days and then transferred to growing conditions of 22 °C and a 16 h light/8 h dark cycle. After seven days, the germination rate was measured and the green cotyledons were scored. The seedlings continued to grow vertically for 20 days inside an artificial climate room (16 h light/8 h dark, 22 °C), and the root length, number of lateral roots and root dry weight were recorded. Each experiment was performed with three replicates for each treatment.

A pot (5 cm × 5 cm) experiment was used to identify the drought resistance of *Arabidopsis* seedlings. Peat/forest soil and vermiculite (1:1 *v/v*) were mixed and used as the substrate. Before drought stress treatment, the seedlings were supplemented with water every three days and Hoagland nutrient solution every two weeks. After the seeds germinated, the seedlings were not watered for two weeks to simulate drought conditions. After the two weeks of drought, the seedlings were rewatered, and the survival rate and relative leaf water content (RWC) of the seedlings were measured after seven days [36].

### 2.4. Measurement of Leaf Stomatal Density, Photosynthesis Rate, Transpiration Rate and Water-Use Efficiency

A leaf surface imprinting method [18] was used to compare sampled leaves of the same age and relative position from WT and transgenic tobacco seedlings. We prepared leaf samples by applying a drop of nail polish on the paraxial surface of the leaf, removing the drying layer with adhesive tape, putting that layer on a glass slide, and putting a drop of water on the imprint on the slide. We observed prepared samples under a Leica CTR6000 electron microscope. Five visual fields were observed and photographed under

a Leica CTR6000 (Leica, Germany) electron microscope. Stomatal lengths and widths were measured by Image J software (National Institutes of Health). Photosynthetic (*P*) and transpiration (*T*) rates were measured by a portable photosynthesis system (LI-COR, Li-6400, LI-COR Environmental, Lincoln, NE, USA). Each measurement was sampled three times. We calculated WUE by *P/T*.

### 2.5. Measurement of MDA and Proline Contents

The content of malondialdehyde was determined according to the method of Chen [37]. We sampled 1.0 g transgenic or WT plant tissue after completing the drought stress treatment on plants. Then, 2 mL of 10% trichloroacetic acid (TCA) and a small amount of quartz sand were added to grind samples. These coarse homogenates received further grinding after adding 8 mL TCA before centrifuging each homogenate at 4000 R/min for 10 min. We collected 2 mL of centrifuged supernatant (2 mL of distilled water was used as a control) and added 2 mL of 0.6% thiobarbituric acid solution. The mixture was then placed into a boiling water bath for 15 min, removed and quickly cooled before centrifuging the mixture. We measured the absorbances of the supernatant at 532, 600 and 450 nm. MDA content was calculated according to the following steps. First, we calculated the content of MDA in the extract: MDA content in the extract ($\mu$mol/L) = 6.45 ($OD_{532} - OD_{600}$) $-$ 0.56$OD_{450}$. Second, we calculated the content of MDA in the fresh tissue sample: MDA content in fresh plant tissue ($\mu$mol/g FW) = MDA content in the extract ($\mu$mol/mL) $\times$ total extract (mL)/fresh weight of plant tissue (g).

The content of proline was determined by the acid ninhydrin method described by Chen [37]. We ground 0.5 g samples of transgenic or WT plant tissue into a homogenate with 5 mL of 3% sulfosalicylic acid before transferring the homogenate into a centrifuge tube to be placed into a boiling water bath for 10 min. Sample tubes were then removed to cool before filtering each solution; the filtrate was collected to continue extracting the proline. From the filtrate, 2 mL was taken to mix with 2 mL glacial acetic acid and 4 mL of 2.5% acid ninhydrin reagent. The mixture was heated in a boiling water bath for 30 min, after which the solution turned red. After cooling, we added 4 mL toluene, shook the mixture for 30 s, allowed any precipitates to settle to the bottom of the tube and transferred the supernatant into a 10 mL centrifuge tube to spin at 3000 R/min for 5 min. The red toluene mixture of proline in the upper layer of each tube was gently transferred into a cuvette, and the toluene solution was used as the blank control. The absorbance values of samples were determined at 520 nm and compared to a standard curve to determine the contents of proline (X) in each sample. The formula used to calculate the contents is: proline content ($\mu$g/g) = {(X $\times$ total amount of extract (mL))/(fresh weight of sample (g) $\times$ amount of extract used in content determination (mL))}.

### 2.6. Gene Expression Analysis

Semi-quantitative RT-PCR and quantitative real-time PCR were used to analyze the overexpression of the *ZmHDZIV13* gene. Total RNA was extracted from the leaves of transgenic and wild *A. thaliana* strains using the RNA Extraction Kit (TIANGEN, Beijing, China). The concentration and quality of RNA were determined by NanoDrop-2000 (Thermo, Waltham, MA, USA). The first strand of cDNA was synthesized by a Reverse Transcription Kit (TIANGEN, Beijing). The real-time fluorescence quantitative PCR was carried out following a two-step method according to the SYBR Premix Ex TaqTM (Takara, Shiga, Japan) kit's manual. PCR reaction conditions were as follows: pre-denaturation at 94 °C for 4 min, followed by 35 cycles of denaturation at 94 °C for 30 s and annealing at 56 °C for 30 s. The relative expression of genes was calculated and analyzed by the $2^{-\Delta\Delta CT}$ method. *His2A* was used as the internal standard to adjust the relative expression levels of transgenic and WT plants [38]. Each treatment was repeated three times.

## 2.7. Statistical Analysis

SPSS 19.0 statistical software (SPSS Inc., Armonk, NY, USA) was used to analyze the data, and Duncan's test was used to assess for significant differences (at $p < 0.01$) between different treatments. The results are presented as the mean ± standard deviation (SD) of three independent biological replicates. Microsoft Excel was used for mapping.

## 3. Results

### 3.1. Informatics Analysis of ZmHDZIV13

The full-length cDNA sequence of *ZmHDZIV13* was cloned from the inbred line Zheng 58 by RT-PCR, resulting in a length of 2097 bp. The full-length DNA sequence of the gene (GenBank accession number: BK008038) was obtained by searching the National Center for Biotechnology Information database with the cDNA sequence as a probe. Sequence alignment showed that the gene contained 10 exons and 9 introns. The DNA sequence of *ZmHDZIV13* was used as a probe to search through a maize genome database (https://maizegdb.org/ (accessed on 28 May 2015)). We found that the *ZmHDZIV13* gene was located near the telomere on chromosome 4. ProtParam analysis showed that the *ZmHDZIV13* gene, with a predicted molecular weight of 7.62 kDa and an isoelectric point of 6.19, encoded 698 amino acids. CD-Search was used to analyze the conserved domain of the *ZmHDZIV13* protein. The results showed that the *ZmHDZIV13* protein had two conserved domains, HD and START, which was consistent with that of the HDG protein (Figure 1). Based on analysis of a phylogenetic tree constructed with 14 maize genes, four *Arabidopsis* genes and one rice gene, *ZmHDZIV13*, were most similar to *AtHDG11*, with approximately 100% homology, followed by *OsROC8*, with 91% homology (Figure 2).

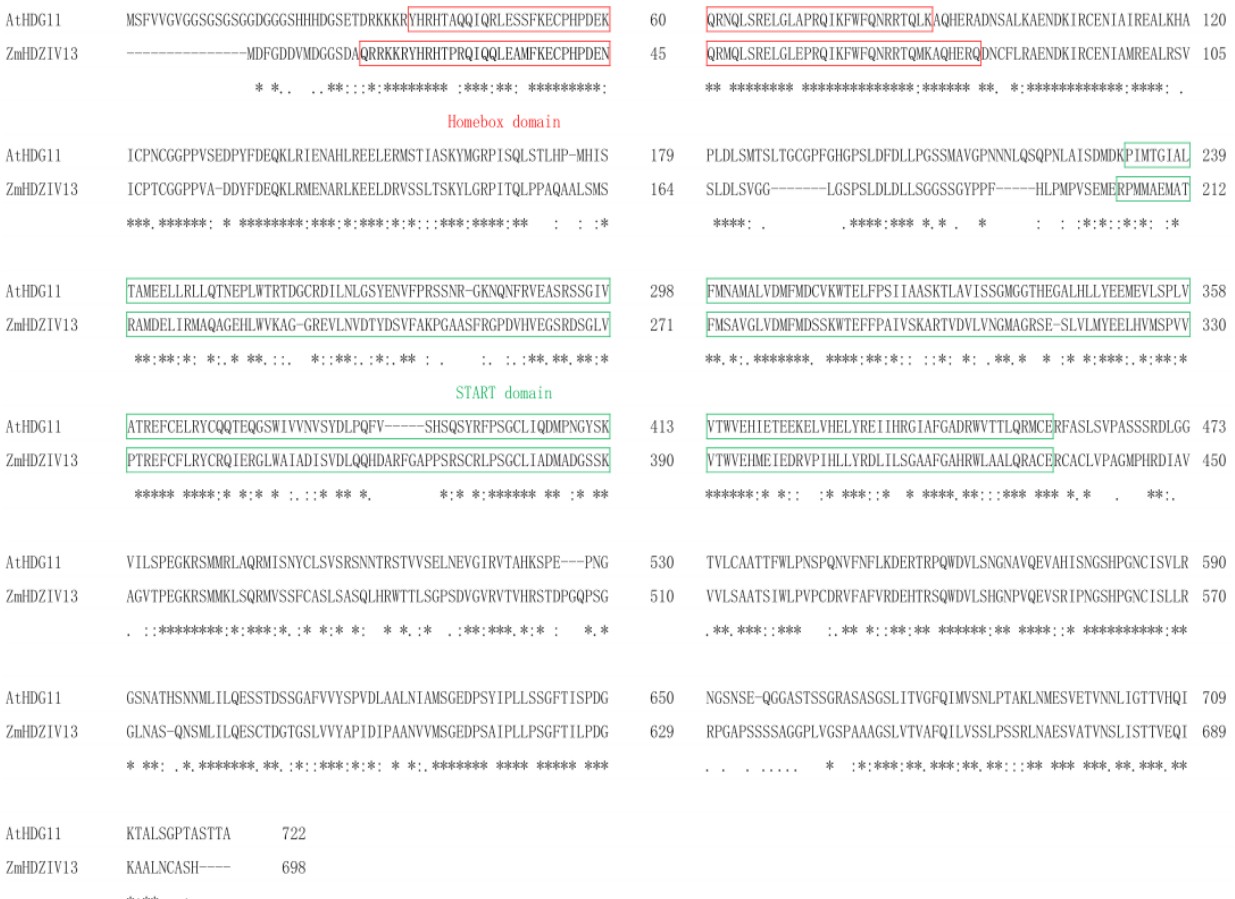

**Figure 1.** Homology analysis of *ZmHDZIV13* and *At HDG11* amino acid sequences.

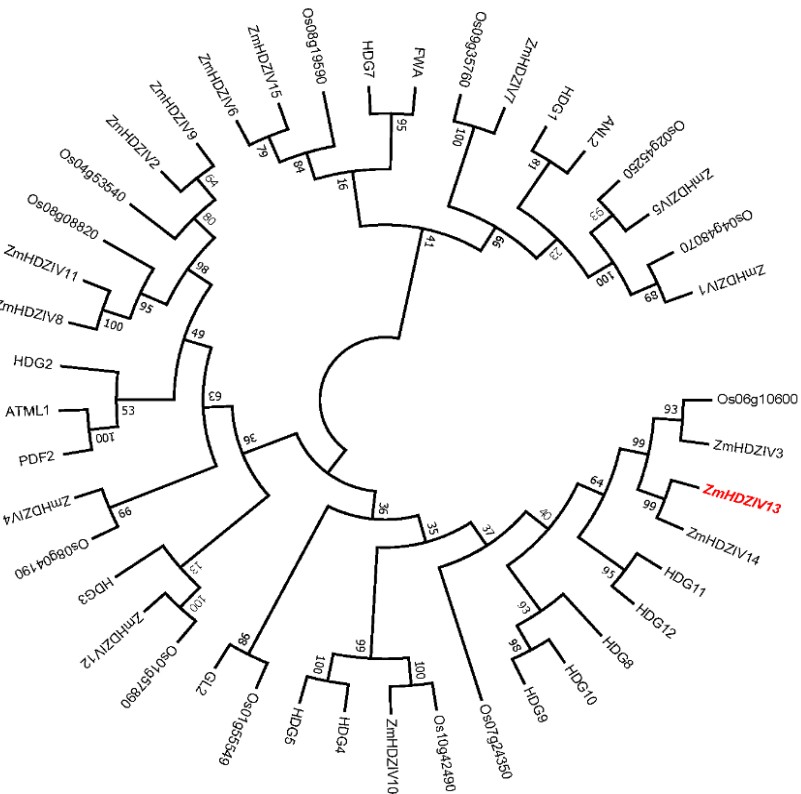

**Figure 2.** Phylogenetic analysis of predicted relationships between *ZmHDZIV13* of maize and *HD-Zip IV* transcription factors of other plant species. Species abbreviations: At, *Arabidopsis thaliana*; Os, *Oryza sativa*; Zm, *Zea mays*.

### 3.2. Transformation of ZmHDZIV13 into Arabidopsis

The results of the transformation of *Agrobacterium* showed that the plant expression vector pCAMBIA3300-35S-*ZmHDZIV13*-bar was successfully introduced into the strain LBA4404. The transgenic line *35S::ZmHDZIV13* was screened by adding 300 μg/L basta in the medium. We obtained 53 homozygous lines of *ZmHDZIV13* in the T3 generation. Expression analysis of *ZmHDZIV13* resulted in 39 resistant plants with specific bands amplified near the 2097 bp size, while no corresponding bands were found in the untransformed plants. The PCR-positive transformation rate of resistant plants was 81.25%, indicating that the exogenous gene was successfully integrated into the *Arabidopsis* genome. The amplification results are shown in Figure S1.

### 3.3. Morphological and Physiological Characteristics in Response to Drought Stress

After seven days of drought treatment, the survival rate and leaf RWC of the T3 generation of overexpression and WT plants were determined. The results showed that the survival rates of transgenic (L3, L7, L25) and WT plants were 100% when given sufficient water. In contrast, the survival rates of WT, L3, L7 and L25 grown in the drought stress condition decreased by 82.0%, 57.63%, 64.41% and 68.38% of that of the WT control, respectively. The survival rates of L3, L7 and L25 were 135.39%, 97.72% and 75.67% higher than those of the WT under stress, respectively. Where water availability was sufficient, the RWCs of WT, L3, L7 and L25 remained above 91%, and the differences among these groups were not obvious (Figure 3a–c). With the temporary drought, the RWCs of leaves of L3, L7 and L25 were reduced. Moreover, RWC of the WT reached the lowest amount, 59.09%, while the RWCs of the *ZmHDZIV13* transgenic plants maintained high relative water content and were 70.89% (L3), 66.88% (L7) and 30.40% (L25) higher than those of the WT (Figure 3b). The results showed that the water-holding capacity of the T3 generation of

plants that overexpressed *ZmHDZIV13* was stronger than that of WT plants subjected to the temporary drought.

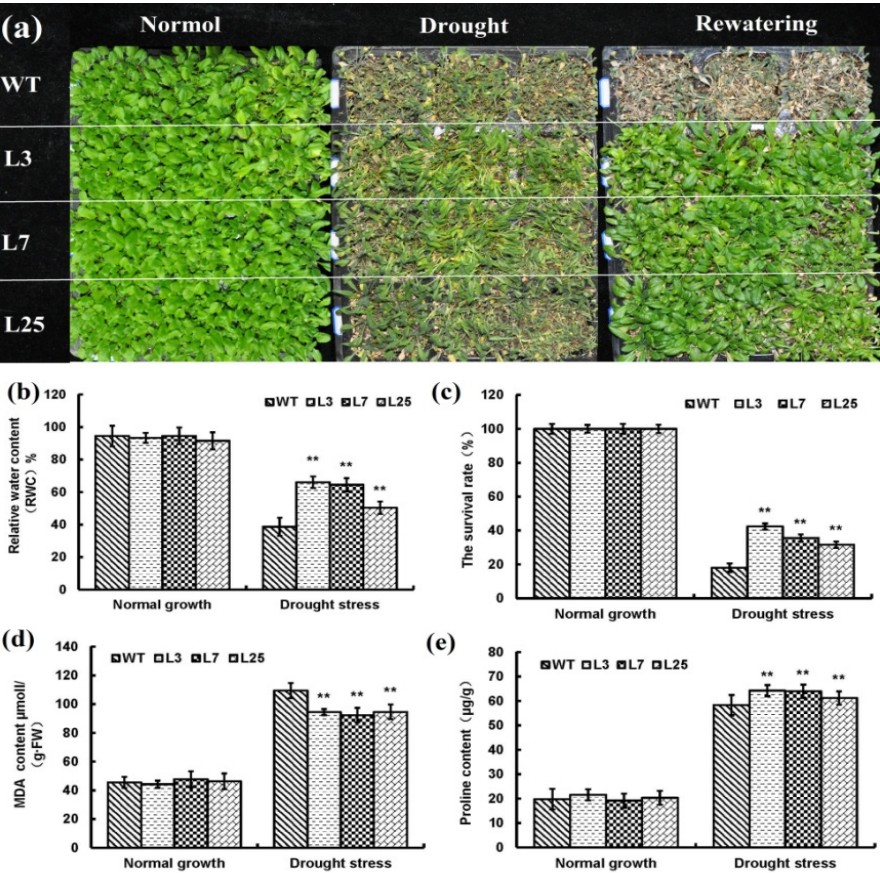

**Figure 3.** The morphological and physiological characteristics of transgenic lines (L3, L7 and L25) and wild-type *Arabidopsis thaliana* under drought stress were studied. ** indicate a significant difference at the 1% level. (**a**) Morphology and development under normal water condition, drought condition and after the restoration of watering of the plant were studied. (**b–e**) RWC, survival rate, MDA content and proline content were measured under normal water and drought stress.

The contents of MDA and free proline are important indexes used to reflect drought resistance of crops. Proline is an osmoprotectant that reduces the osmotic potential of cells in response to drought stress. MDA is a product of plant cell membrane lipid peroxidation. Drought stress causes plant cells to lose water and eventually damages the membrane system, observed in the ruptures of the vacuolar membrane and damages to the lipid membrane structure. In our experiment, the contents of MDA and proline were determined from WT, L3, L7 and L25 plants exposed to drought stress. The results showed that the contents of MDA and proline in the non-transformed and transformed lines were similar to the corresponding contents in the regularly irrigated group. After seven days of water deprivation, compared with WT under drought stress, the MDA contents of L3, L7 and L25 significantly decreased by 13.67%, 15.63% and 13.60%, respectively, and the proline contents of L3, L7 and L25 significantly increased by 10.14%, 9.74% and 4.99%, respectively, from those of their non-deprived groups (Figure 3d,e). The results showed that the physiological indexes related to drought resistance observed from the transformed plant leaves were improved in response to the temporary drought treatment.

### 3.4. Morphological Characteristics in Response to Different ABA Levels

ABA signaling stimulates plant responses to various stress factors by regulating the expression of stress- and ABA-responsive genes and the closure of leaf stomata [33]. In order

to determine whether the *ZmHDZIV13* gene improves drought resistance in transgenic *Arabidopsis* through the ABA signaling pathway, the phenotypes of WT and transgenic *A. thaliana* seedlings grown on MS medium containing 0.6 μM ABA were observed and compared. It was found that the growth phenotypes (seedling length and root dry weight) of transgenic L3, L7 and L25 were significantly higher than those of WT in MS medium without ABA (Figure 4c,e). In the medium containing 0.6 μM ABA, the growth of WT and transgenic *Arabidopsis* seedlings was significantly inhibited, and the degree of inhibition was significantly greater in transgenic than in WT plants. These results indicate that overexpression of *ZmHDZIV13* can significantly increase the sensitivity of transgenic plants to ABA.

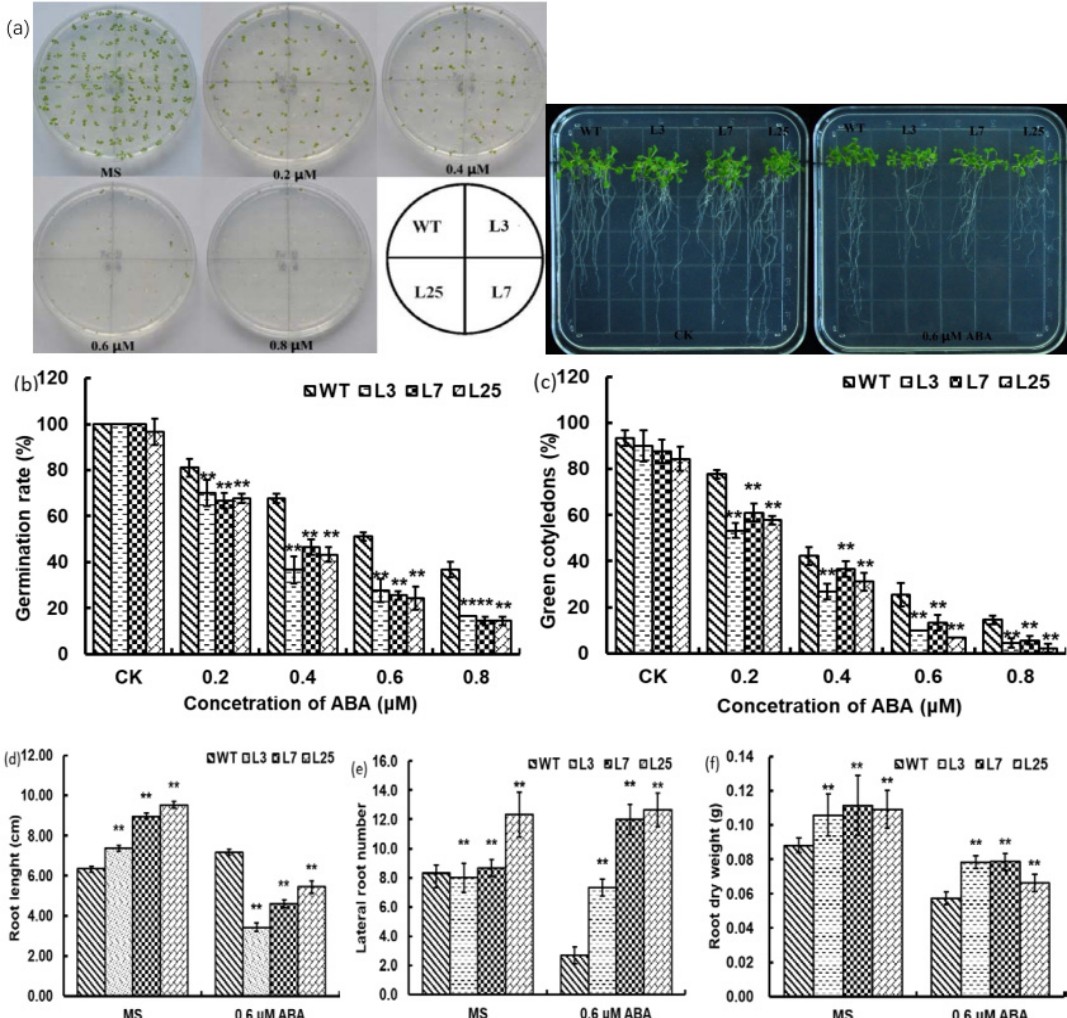

**Figure 4.** Morphological characteristics of transgenic (L3, L7 and L25) and WT lines at different ABA levels (CK: 0.2, 0.4, 0.6 and 0.8 μm ABA). ** indicate a significant difference at the 1% level. (**a**) The germination and root morphology of plants treated with ABA were studied. (**b**) Germination rate (%). (**c**) Green cotyledons (%). (**d**) The root length was measured on the 20th day. (**e**) The number of lateral roots was measured on the 20th day. (**f**) Root dry weight was measured on the 20th day (**g**).

### 3.5. Morphological Characteristics in Response to NaCl Treatments and Osmotic Stress in Arabidopsis

The WT *Arabidopsis* and *35S::ZmHDZIV13* transgenic lines (L3, L7, L25) were subjected to osmotic stress tests on a medium supplemented with NaCl and mannitol to determine potential interactive effects of genetics and environmental conditions. The results showed that the growth of seedlings was inhibited under both stress conditions, and the inhibition

increased with the increase in NaCl/mannitol concentration, but the germination rate and green cotyledon emergence rate of WT were lower than those of *35S::ZmHDZIV13* transgenic lines (Figures 5b,c and 6b,c). These results indicate that *ZmHDZIV13* can improve germination and cotyledon emergence rates of transgenic plants under osmotic and salt stresses.

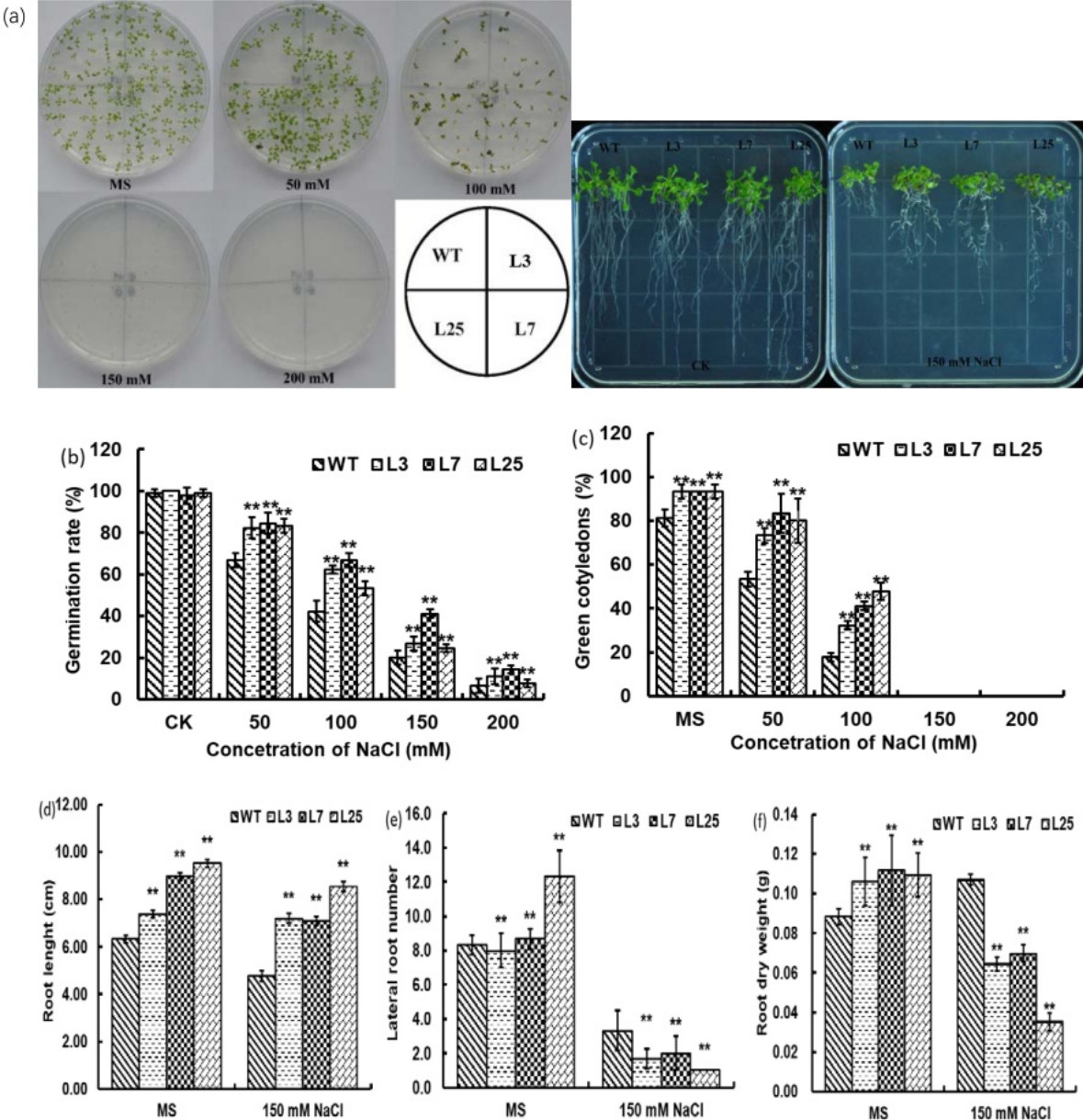

**Figure 5.** The morphological characteristics of transgenic *Arabidopsis thaliana* were different under the NaCl treatments (CK, 50, 100, 150 and 200 mM). ** indicate a significant difference at the 1% level. (**a**) The germination and root morphology of plants treated with NaCl were studied. (**b**) Germination rate (%). (**c**) Green cotyledons (%). (**d**) The root length was measured on the 20th day. (**e**) The number of lateral roots was measured on the 20th day. (**f**) Root dry weight was measured on the 20th day (g).

Root length, number of lateral roots and root dry weight of *35S::ZmHDZIV13* and WT *Arabidopsis* seedlings under NaCl and mannitol stresses were measured or counted. Under NaCl and mannitol stresses, the number of lateral roots from *ZmHDZIV13* was higher than that of WT (Figures 5e and 6e) and the transgenic *35S::ZmHDZIV13* seedlings showed strong osmotic resistance. With the addition of 150 mM NaCl, the average root length

and root dry weight of transgenic *Arabidopsis* seedlings decreased by 57.59% and 4.59%, respectively, while the number of lateral roots increased by 10.34%. The root length, root dry weight and number of lateral roots of WT decreased by 80.95%, 35.23% and 67.47%, respectively (Figure 5d–f). Due to treatment with 150 mM mannitol, the average root length and root dry weight of transgenic *Arabidopsis* seedlings decreased by 11.97% and 11.36%, respectively, while the number of lateral roots increased by 88.50%. The root length, root dry weight and number of lateral roots of WT decreased by 11.94%, 54.55% and 72.29%, respectively (Figure 6d–f). The results showed that *35S::ZmHDZIV13* seedlings exhibited stronger tolerance to the drought stress, and this might be due to the increased number of lateral roots.

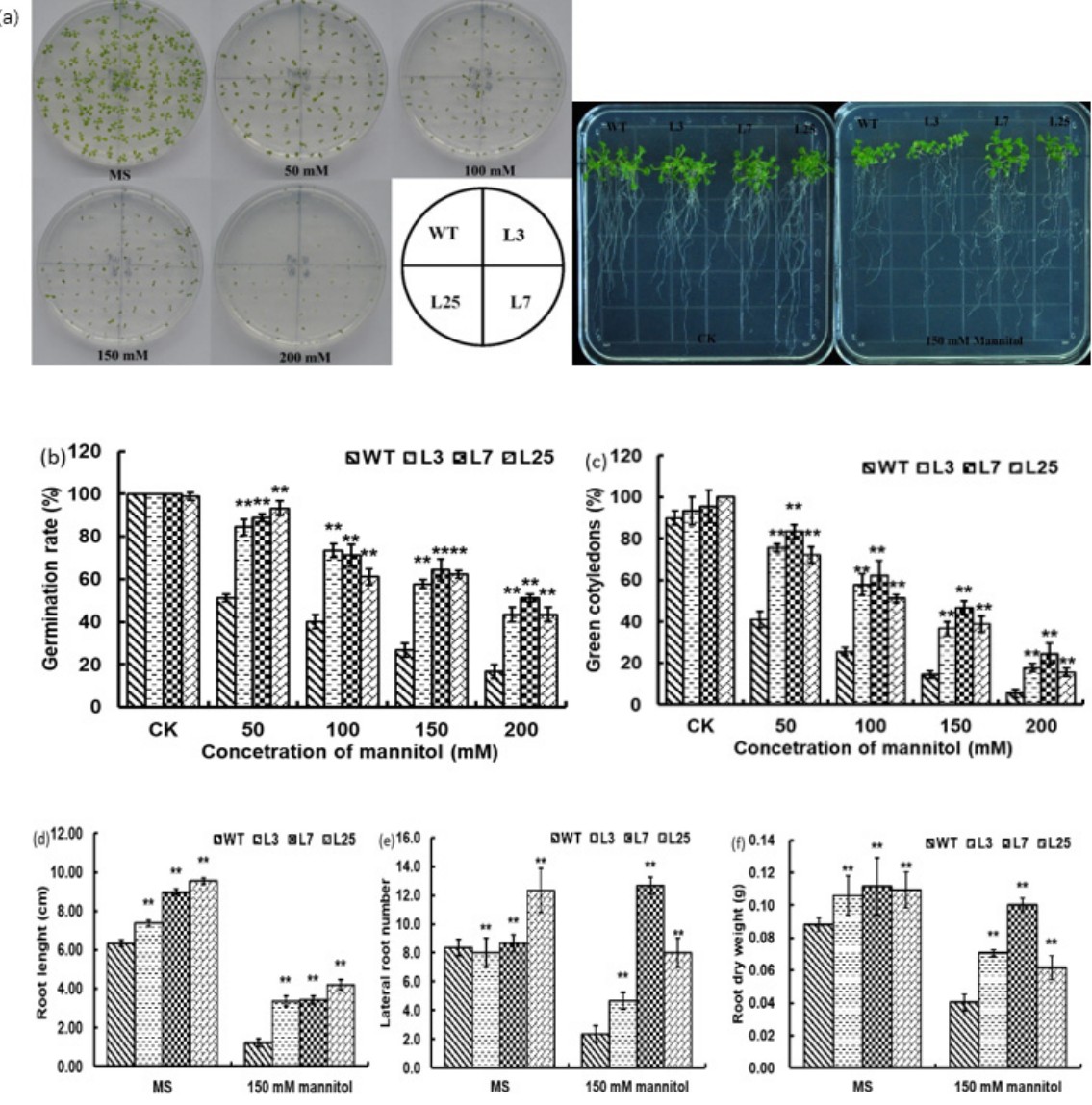

**Figure 6.** The morphological characteristics of transgenic *Arabidopsis thaliana* were different under the mannitol treatments (CK, 50, 100, 150 and 200 mM). ** indicate a significant difference at the 1% level. (**a**) The germination and root morphology of plants treated with mannitol were studied. (**b**) Germination rate (%). (**c**) Green cotyledons (%). (**d**) The root length was measured on the 20th day. (**e**) The number of lateral roots was measured on the 20th day. (**f**) Root dry weight was measured on the 20th day (g).

### 3.6. Relative Expression Levels of Stress Response Genes under Drought Stress

In order to elucidate the molecular mechanism of drought tolerance of the *ZmHDZIV13* gene, the transcription levels of six drought-related genes were compared by quantitative PCR. The results showed that ABA and drought stress-inducible genes *(P5CS1, RD22, RD29B, RD29A, ERD1, NCED3)* were highly expressed in dehydrated plants, and the expression levels of drought genes in transgenic plants were significantly higher than those in WT plants (Figure 7). For example, the relative expression levels of *RD29B* in *ZmHDZIV13* transgenic lines L3, L7 and L25 were 127.70, 126.41 and 153.05, respectively, on the 6th day of dehydration (Figure 7d). In addition, we found that drought could induce the expression of *ERD1* (Figure 7f), but there was no significant difference in expression between transgenic and WT *Arabidopsis*. Before the drought treatment, with the exception of *RD29B* (Figure 7d), the expression levels of all other genes in transgenic plants were lower than those in WT.

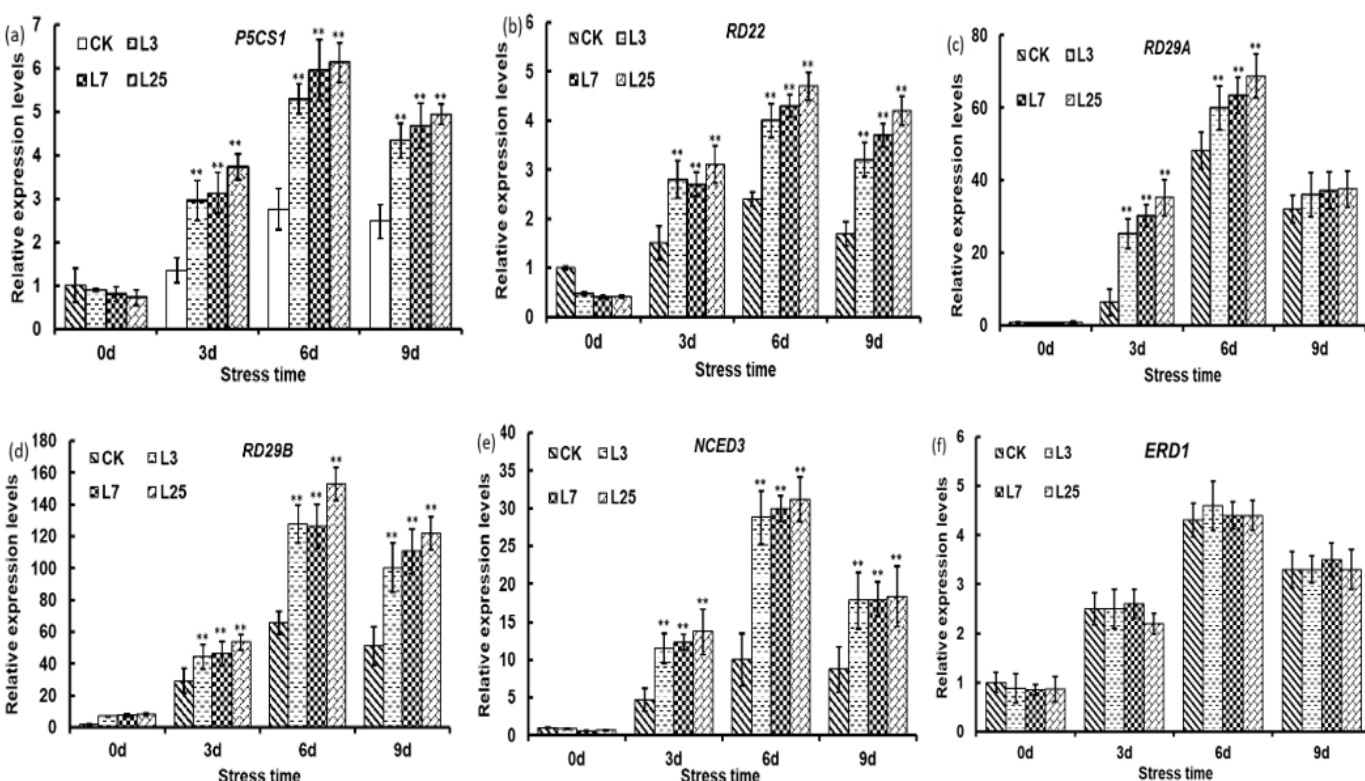

**Figure 7.** Relative expression levels of six stress response genes (*P5CS1*, *RD22*, *RD29B*, *RAB18*, *NCED3* and *ERD1*) in *Arabidopsis* lines (L3, L7 and L25) at 0, 3, 6 and 9 days after drought stress. ** indicate a significant difference at the 1% level. (**a**) Relative expression levels of *P5CS1* in *Arabidopsis* lines (L3, L7 and L25) at 0, 3, 6 and 9 days after drought stress. (**b**) Relative expression levels of *RD22* in *Arabidopsis* lines (L3, L7 and L25) at 0, 3, 6 and 9 days after drought stress. (**c**) Relative expression levels of *RD29A* in *Arabidopsis* lines (L3, L7 and L25) at 0, 3, 6 and 9 days after drought stress. (**d**) Relative expression levels of *RD29B* in *Arabidopsis* lines (L3, L7 and L25) at 0, 3, 6 and 9 days after drought stress. (**e**) Relative expression levels of *NCED3* in *Arabidopsis* lines (L3, L7 and L25) at 0, 3, 6 and 9 days after drought stress. (**f**) Relative expression levels of *ERD1* in *Arabidopsis* lines (L3, L7 and L25) at 0, 3, 6 and 9 days after drought stress.

### 3.7. Leaf Density and Water-Use Efficiency in Transgenic Tobacco

Heterologous reproduction experiments were carried out with transgenic tobacco. The genetic structure pCAMBIA3300-*35S-ZmHDZIV13-bar* was transferred into tobacco and further studied using three independents overexpression lines T2 homozygous lines (L7, L10 and L17) (Figure S2). The results showed that stomatal density of transgenic *ZmHDZIV13* overexpression lines in L10 and L17, but not in L7, were lower than in WT

lines (Figure 8a,e), while the size of a single stomate of each transgenic line was greater than that of WT lines (Figure 8f). The lower stomatal density may be related to the greater size of epidermal cells (Figure 8b), and the decrease in stomatal density may help to reduce water loss from the leaves of transgenic plants. The photosynthetic rates were significantly greater, transpiration rates were significantly lower (Figure 8b,c), and WUEs were significantly higher in transgenic than in WT lines (Figure 8d).

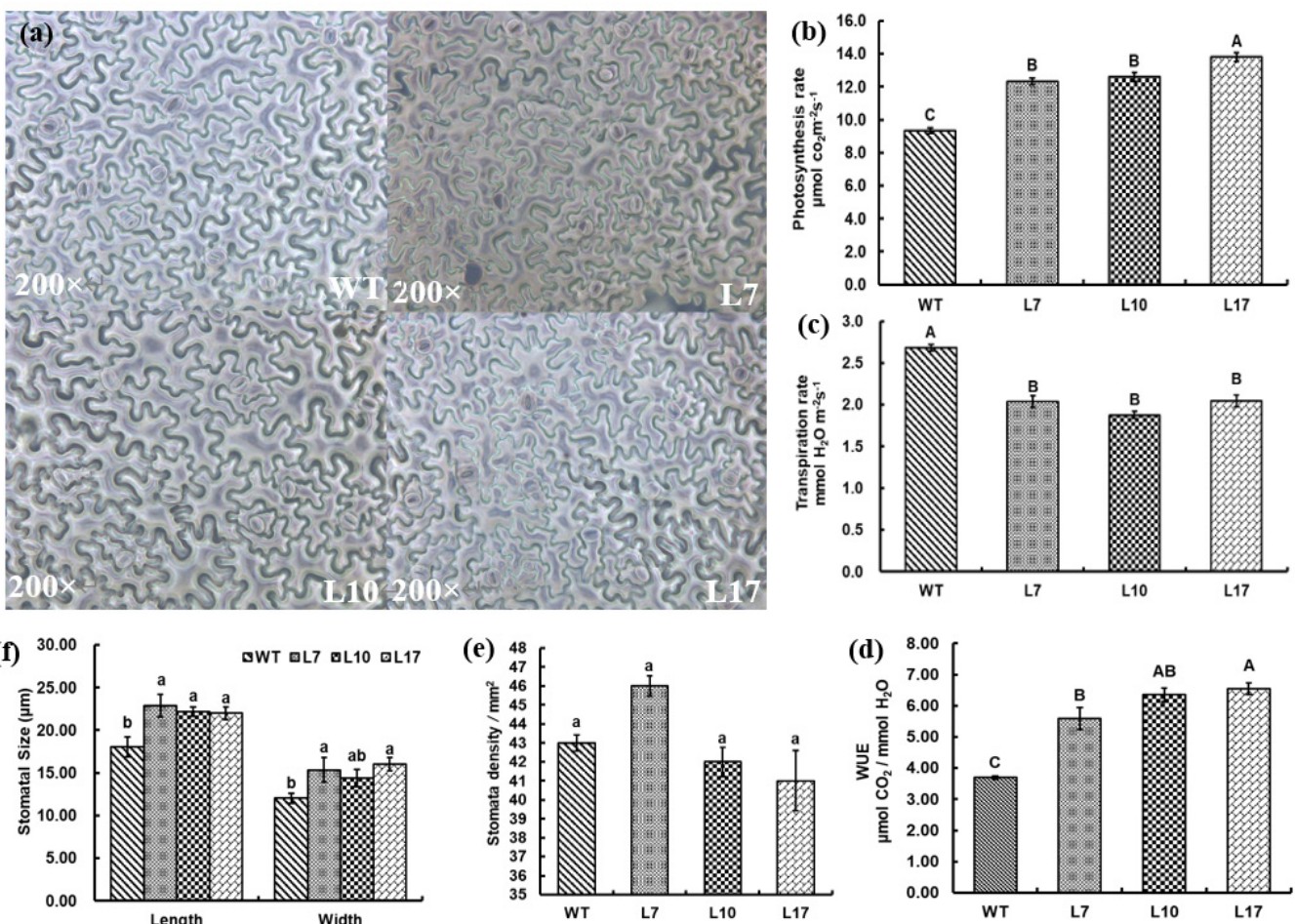

**Figure 8.** Stomatal density (**a**,**e**), stomatal size (**f**), photosynthesis rate (**b**), transpiration rate (**c**) and water-use efficiency (WUE) (**d**) in transgenic (L7, L10, and L17) and WT tobacco lines. Lowercase letters indicate a significant difference at the 5% level. Capital letters indicate a significant difference at the 1% level.

## 4. Discussion

Current studies have shown that the induced expression of transcription factors, such as *Os MYB3R-2*, *CBF1*, *Os SDIR1*, *ABF2* and *SlHZ24*, in response to stress [39–42], can improve the resistance of plants to stress, and increasing evidence shows that HD-ZIP family proteins participate in plant response to stress [43]. HD-Zip showed a universal induction level under drought stress. The drought stress response mechanism involved in HD-Zip has been widely studied in dicotyledons and monocotyledons [44]. The HD-Zip IV subfamily of transcription factors are mostly and specifically expressed in the epidermal tissues of plant organs, and they are mainly involved in the accumulation of anthocyanins, epidermal differentiation and trichome formation on plant organs [16,45]. There are 16 members of the IV subfamily in *Arabidopsis* [45]. Studies have found that the *ATHB10* gene mainly functions in the root hair, epidermal hair and seed coat [46]. Vernoud et al. [8] found that the *OCL4* gene of maize is mainly expressed in the leaf epidermis. Overexpression of this gene will

inhibit the development of transgenic *Arabidopsis* leaf epidermal trichomes, while silencing the expression of this gene will cause abnormal phenotypes in epidermal hair differentiation in plants. Research by Li et al. [47] showed that overexpression of *HGD3* causes anthers to remain closed and leads to male sterility. In our study, we cloned the full-length cDNA of *ZmHDZIV13* (an HD-Zip IV gene), analyzed the effects of endogenous ABA, NaCl and mannitol on *ZmHDZIV13* transgenic *Arabidopsis* and tobacco lines, and evaluated how *ZmHDZIV13* regulates downstream genes in transgenic *Arabidopsis* and tobacco.

ABA signaling stimulates plant responses to various stress factors by regulating the expression of stress and ABA-responsive genes and the closure of leaf stomata [33]. Studies have shown that a similar HD-Zip I gene, *Hahb-4*, was expressed in sunflower (*Helianthus annuus* L.) in the early stage of development and could be induced by abiotic stress from water deficiency or by ABA [48]. Studies on the HD-Zip I gene in rice showed that overexpression of *ZMHDZ10* enhanced drought and salt tolerance, but increased sensitivity to ABA [49]. In addition, previous studies have found that overexpression of *AtEDT1/HDG11* can lead to ABA hypersensitivity, thus inhibiting germination and inducing stomatal closure [30]. Overexpression of *ZmHDZ4* and *ZmHDZ10* in maize inhibited the germination rate in MS medium containing ABA, and the seedling height and root length were higher than those of the wild-type, indicating that both showed high sensitivity to ABA and may affect ABA signaling [49,50]. In this study, we found that overexpression of *ZmHDZIV13* can significantly improve the sensitivity of transgenic *Arabidopsis* to exogenous ABA, which indicates that *ZmHDZIV13* may play a positive role in regulating plant stress response through an ABA-dependent signal transduction pathway. *Oshox22*, *Athb7*, *Athb12* and *Zmhdz10* are members of the HD-Zip I family [51], and they have been reported to regulate plant responses to various stress factors by participating in the ABA signaling pathway. Similar to the genes of HD-Zip I, *ZmHDZIV13* was not only induced by drought, but also by exogenous ABA, indicating that *ZmHDZIV13* may be directly or indirectly involved in the ABA signaling pathway and may regulate the adaptability of plants to drought through the ABA signaling pathway.

In the process of plant evolution, in order to adapt to the changing external environment and to maintain their growth, plants have formed a set of complex and rigorous stress response mechanisms. When plants are stimulated by external stress, the content of reactive oxygen species within the plants accumulates, resulting in lipid peroxidation and peroxidation stress, which are not conducive to the normal growth of plants. The HD-Zip gene can increase the activity of antioxidant enzymes and the accumulation of some soluble organic substances [44]. Drought tolerance in transgenic maize containing the *ZmHDZIV13* gene was investigated. After 7 days of drought, the untransformed plants exhibited serious damage; for example, their leaves showed different degrees of wilting and curling. However, the drought tolerance of transformed maize plants was significantly enhanced and they maintained normal growth. Malondialdehyde is commonly used as an indicator of the degree of lipid peroxidation [52]. In response to the drought stress, MDA contents in transgenic plants overexpressing *ZmHDZIV13* were significantly lower than in the control group, indicating that overexpression of *ZmHDZIV13* can reduce the degree of membrane lipid peroxidation in transgenic plants and reduce the degree of cell membrane damage. Free proline in plants has important functions. In plants under stress, free proline can stabilize subcellular structure, maintain intercellular permeability and reduce cell damage by accumulating proline [53]. Our results showed that the free proline content in transgenic plants was significantly higher than in the WT group subjected to drought. Meanwhile, the accumulation of proline reduced the water potential in *Arabidopsis* and inhibited the loss of water from cells, which is consistent with the RWCs observed in leaves under drought stress. The water content of leaves in the transgenic group was significantly higher than in WT. The results also showed that cell membrane damage in transgenic plants under stress was less than the damage in the WT. Moreover, resistance to membrane lipid peroxidation was significantly enhanced, which improved the growth and development of individuals and further protected these transgenic *Arabidopsis* plants from

the abiotic stress. These results indicate that *ZmHDZIV13* is overexpressed in maize inbred lines, which helps maintain plasma membrane stability and improve antioxidant capacity.

The research shows that overexpression of stress-inducible transcription factors usually improves the resistance of plants to various stresses by regulating the expression of downstream stress response genes [54]; thus, it is necessary to identify the target genes downstream of *ZmHDZIV13*. In this study, we found, through homologous alignment, the genes that play key roles in the drought signaling pathway in *A. thaliana* (*ERD1*, *RD29A*, *P5CS1*, *RD22*, *NCED3* and *RD29B*). The *RD29B*, *RD22* and *RD29A* genes have been shown to be primarily involved in drought stress and ABA response [55] and can improve plant tolerance to stress, such as dehydration, and ABA induction [2,49,56]. *NCED3* improves drought resistance and salt tolerance by regulating ABA synthesis in plants [2]. *ERD1* is an early drought-inducible gene. The induced expression of *P5CS1* can lead to the accumulation of proline, which results in strong stress resistance in the *ZmHDZIV13*-overexpressing transgenic plants. The expression of stress-related genes in transgenic *A. thaliana* was examined by fluorescence quantitative PCR. The results showed that the expression levels of all genes, except *RD29B*, in transgenic plants were lower than those in WT plants when given sufficient water. In response to the drought stress treatment, the expression levels of all genes were up-regulated to varying degrees, and the expression levels of all genes in transgenic seedlings were significantly higher than those in WT seedlings. These results indicate that overexpression of the *ZmHDZIV13* gene plays an important role in increasing the expression of six marker genes in transgenic plants, and that *ZmHDZIV13* may participate in an ABA-dependent signal transduction pathway. Zhu et al. [30] also showed that *AtEDT1/HDG11* enhanced abiotic stress resistance in Chinese kale through auxin and ABA-mediated signal transduction. Therefore, the activation of these stress resistance genes plays a crucial role in improving the stress resistance of *ZmHDZIV13* transgenic plants. In conclusion, *ZmHDZIV13* may be an important upstream gene regulating ABA biosynthesis and signal transduction.

## 5. Conclusions

In this study, we isolated *ZmHDZIV13* from the HD-Zip IV transcription factor family in maize and characterized its role in drought and salt stress and its sensitivity to ABA. The results show that the transgenic *A. thaliana* had obvious drought resistance compared with the control WT under drought stress. The MDA content of *ZmHDZIV13* transgenic plants was lower than that of the control, and the relative water content and proline content were significantly higher than those of the control. After drought relief, the expression levels of *P5CS1*, *RD22*, *RD29B*, *RD29A*, *NCED3* and *ERD1* in transgenic *A. thaliana* were up-regulated. This indicates that *ZmHDZIV13* can regulate the drought response of plants by regulating the ABA-dependent signaling pathway. These results provide a basis for further study of the function of the *ZmHDZIV13* gene and the creation of transgenic drought-resistant materials in the future.

**Supplementary Materials:** The following supporting information can be downloaded at: https://www.mdpi.com/article/10.3390/agronomy12102378/s1, Figure S1: RT-PCR of the *35S::ZmHDZIV13* transgenic and WT *Arabidopsis* lines. M, DNA Marker D5000.CK+, positive control. CK-, negative control. 1~10, resistant plants. Figure S2: RT-PCR of the *35S::ZmHDZIV13* transgenic and WT tobacco. M, DNA marker VII. CK+, positive control. CK-, negative control. 1~10, *ZmHDZIV13* resistant plants.

**Author Contributions:** Y.P. designed the experiments. P.F. and H.Y. performed the experiments. Y.P., P.F., F.W. and X.J. analyzed the data. Y.P., P.F. and F.W. wrote the manuscript. All authors have read and agreed to the published version of the manuscript.

**Funding:** This research was supported by the research program sponsored by the Gansu Provincial Higher Education Industry Support Plan (2022CYZC-46), the Gansu Agricultural University Fuxi Talent Plan (GAUFX-02Y09) and the Industrial Support Project of Colleges and Universities in Gansu Province (2021CYZC-12).

**Institutional Review Board Statement:** Not applicable.

**Informed Consent Statement:** Not applicable.

**Data Availability Statement:** Not applicable.

**Conflicts of Interest:** The authors declare that they have no conflict of interest.

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
