# Peer review of "Heterologous Overexpression of ZmHDZIV13 Enhanced Drought and Salt Tolerance in Arabidopsis and Tobacco"

_agronomy, doi:10.3390/agronomy12102378_

Round 1
Reviewer 1 Report
Comments on agronomy- 1866120-manuscript:
This manuscript presents a potentially interesting and significant question, that is ZmHDZIV13, as a stress-responsive 100 transcription factor, plays a role in the positive regulation of abiotic stress tolerance and 101 is of great significance for improving maize stress resistance and enriching resources 102 based on maize resistance genes. The authors provided a reliable reference of about cloning the HD-Zip-IV 17 gene ZmHDZIV13 and identified its function in the stress response. They apply apparently scientific sound approaches and analyses (I use "apparently" only because some details are missing, it is not a judgement of scientific quality).
· Introduction don’t provide sufficient background and must be included recent and relevant references
· The lack of recent references and an inadequate presentation of the results significantly altered the interest and scope of this work.
· Discussion needs more and more amendments and, have to be more stand on the genetic evidences. Also, the discussion needs more explanation’s for significant results.
L 160: add the reference of Measurement of MDA and proline contents
L 161-173: Not italic
L190: add the reference of Gene expression analysis
L219: the figure is not clear
L460-461: No discussion in Conclusion part
L 474: References:
DOI of all references must be add
Journal name must be italic
Reviewer 2 Report
The manuscript entitled “Heterologous overexpression of ZmHDZIV13 enhanced drought and salt tolerance in Arabidopsis and tobacco” explained the effect of maize HD-Zip IV transcription factor, ZmHDZIV13 overexpression on abiotic stress in Arabidopsis and tobacco. Overall, this study has significant importance in understanding the function of the HD-Zip IV transcription factor in maize abiotic stress tolerance. I have some major concerns before considering for publication in Agronomy journal:
1. In the abstract author mention that they have analyzed the phylogenetic tree of the HD-Zip IV transcription factor, but there is no data available in the result section. Author should provide all the bioinformatics data that they have explained in the result section.
2. In line 225, Expression analysis of ZmHDZIV13 resulted in 39 resistant plants with specific bands amplified near the 2097-bp size, but no figure has been provided by the author. Author should supplement all the PCR results.
3. In lines 243-245, the RWC of the WT reached the lowest amount, 59.09%, while the RWCs of the ZmHDZIV13-transgenic plants basically remained above 50% and were 70.89% (L3), 66.88% (L7) and 30.40% (L25) higher than that of the WT, but RWC of L25 lines was 30.40% than how it become above 50% and higher than WT?
4. In lines 257-261, The MDA and proline content of transgenic plants presented as compared to WT or non-stressed plants? Author should clarify and rewrite their results.
5. There is no information about statistical analysis in all figures. Author must need to incorporate the statistical method that they have used either in the method section or in every figure.
6. In figure 4, why author only provided numerical data, why not illustrative data? Author must need to provide all the pictographic data along with numerical data for more reliability of their findings.
7. In lines 275-276, There were no significant differences in seedling and root lengths between WT and the transgenic L3, L7 and L25 plants grown in MS medium without ABA (Fig. 4). But I have seen clear significant differences in root length between WT and transgenic plants? Please explain.
8. In lines 288-291, The results showed that the germination rate of transgenic plants increased with the increase of NaCl concentrations, but the germination rate of transgenic plants was lower than that of WT plants (Fig. 5). According to Fig 5, Germination rate is significantly increased in all transgenics plants compare to WT when exposed to different NaCl concentrations, but you said germination rate of transgenic plants was lower than that of WT plants, please explain?
9. In figure 5, why author only provided numerical data, why not illustrative data? Author must need to provide all the pictographic data along with numerical data for more reliability of their findings.
10. In lines 297-311, Is this the result of transgenic plants compared to WT or non-treated plants? Please clarify and rewrite for better understanding.
11. In figure 6, why author only provided numerical data, why not illustrative data? Author must need to provide all the pictographic data along with numerical data for more reliability of their findings.
12. There is no statistical analysis in Fig. 7. Author should provide it. Why author only checked the expression of drought-responsive genes and not salt stress-related marker genes, as they found significant physiological differences in transgenic plants after salt stress?
13. In Fig. 8, there is no result of transformation confirmation of ZmHDZIV13 in T2 homozygous lines such as RT-PCR. Do the results shown in Fig. 8 after stress treatment? If yes then need to state it properly in the result.
I am not satisfied with how the author wrote their result section. In some cases, it’s really difficult to understand. Author should re-write this section with clarity. All the figures should be numbered like Fig. 1 a b c d and need to explain and cite every single figure in the result section. There also some are English writing errors that should need to be considered.
14. In line 363, analyzed the effects of endogenous ABA, NaCl, and mannitol on ZmHDZIV13-transgenic Arabidopsis and tobacco lines. Author also checked the effect of drought stress but it's only for Arabidopsis transgenic lines and not all for tobacco lines?
15. In lines 374-388, the author explained the same result repeatedly and very difficult to understand what they really want to clarify here. The results also showed that the expression of ABA-responsive genes in transgenic plants was activated due to stress induction. Which ABA-responsive genes were activated need to explain with references?
16. In lines 396-397, In contrast, drought tolerance in transformed maize plants was significantly enhanced and these plants maintained normal growth and produced healthy-looking phenotypes. Is it transformed maize plant?
17. In line 418, Remove or change the word “reportedly”.
18. In conclusion, ZmHDZIV13 responds to stress by not only regulating gene expression but also some physiological changes. At this stage of the study author still has not confirmed that ZmHDZIV13 regulates stress-related genes, so I think it is not reasonable to use “ZmHDZIV13 regulates stress-related genes”. Author should summarize all results properly and state their future research direction.
Round 2
Reviewer 1 Report
No comments
Reviewer 2 Report
The author addressed most of the concerns in the revised manuscript but still have some need to address before acceptance,
1. In Fig. 2, provide all the HD-Zip IV transcription factor genes from Arabidopsis thaliana and Oryza sativa and reconstruct the phylogenetic tree.
2. In Fig. 2 legend, the author said that ZmHDZIV13 of foxtail millet (Sorghum bicolor). Is ZmHDZIV13 isolated from foxtail millet?
3. Fig. 3 and Fig. 9 can place as supplementary figures.
4. In line 209, provide the P value for Duncan’s test.
